# Forensic Prompting with Dual-Action Policy Optimization for Vision-Language Forgery Detection and Localization

**Ye Zhu** [1]  **Ai Zhao** [1]  **Jinwei Wang** [2]

## Abstract

Image forgery is rapidly evolving, rendering forensic traces increasingly subtle and readily attenuated by post-processing. Although vision-language prompting can inject priors, open-ended LLM-generated prompts are difficult to constrain, and naive language descriptions can introduce semantic perturbations. To address these challenges, we propose *Forensic Prompting with Dual-Action policy optimization* (FPDA) for vision-language forgery detection and localization, where the *Forensic Prompting Module* (FPM) constructs a structured and reproducible forensic prompt bank and supports optional text input as a reliability-aware cue for stable conditioning. Moreover, *Dual-Action Policy Optimization* (DAPO) is applied to learn sample-adaptive evidence usage by routing forensic prompts and scheduling localization refinement on a per-image basis, stabilizing discriminative cues and improving mask spatial consistency. Extensive experiments are conducted on multiple public datasets covering manual manipulations, diffusion content, face forgeries, and text-enabled settings, demonstrating favorable detection and localization performance over representative state-of-the-art methods under comparable evaluation protocols.

## 1. Introduction

Image forensics now faces a broader spectrum of manipulations, ranging from classic manual edits (e.g., splicing and copy-move) to diffusion-driven synthesis and instruction-based generative editing (Ho et al., 2020; Rombach et al., 2022; Brooks et al., 2023). As a consequence, forensic traces become increasingly subtle and highly non-stationary:

artifacts evolve with the generator and the editing mechanism, and are often attenuated or even erased by common post-processing (e.g., compression and resizing), which undermines the stability of both detection and localization (Yu et al., 2024; Tan et al., 2024; Yan et al., 2024; Kashiani et al., 2025). To stabilize such unstable evidence, recent vision-language forensic methods leverage textual conditioning (e.g., CLIP-style prompt learning or multimodal large models) as semantic constraints (Li et al., 2024; Guo et al., 2025; Huang et al., 2025). However, open-ended LLM-generated prompts are hard to control and reproduce (and may add inference overhead), external descriptions are often missing or noisy, and uncalibrated fusion can introduce prompt-side semantic perturbations that weaken the alignment between image-level evidence and pixel-level cues, leading to localization drift.

Beyond image-level detection, modern benchmarks increasingly demand pixel-level localization that is consistent with global predictions in terms of spatial attribution, yet localization is particularly fragile under complex backgrounds and subtle edits (Kong et al., 2025). When cues are weak or heavily post-processed, the appropriate refinement strength becomes highly sample-dependent: a fixed pipeline may over-smooth boundaries in some cases while failing to suppress spatial drift in others. This motivates a sample-adaptive mechanism that jointly determines which forensic evidence should be emphasized and how the selected evidence should guide localization refinement.

Motivated by these challenges, we propose *Forensic Prompting with Dual-Action policy optimization* (FPDA), a vision-language framework for joint forgery detection and localization. Instead of treating localization as an isolated auxiliary output, FPDA couples localization with image-level prediction so that the predicted mask remains consistent with the global forgery decision, as illustrated in Figure 1.

FPDA comprises two coordinated components: **(i)** the *Forensic Prompting Module* (FPM) constructs a structured and reproducible forensic prompt bank and applies reliability-gated fusion to provide stable and controllable conditioning against missing/noisy descriptions and prompt-side semantic perturbations; **(ii)** *Dual-Action Policy Optimization* (DAPO) learns two discrete actions, text-side

---

[1]Hebei University of Technology, Tianjin, China [2]Nankai University, Tianjin, China. Correspondence to: Jinwei Wang <vjwei2004@163.com>.

*Proceedings of the 43rd International Conference on Machine Learning*, Seoul, South Korea. PMLR 306, 2026. Copyright 2026 by the author(s).

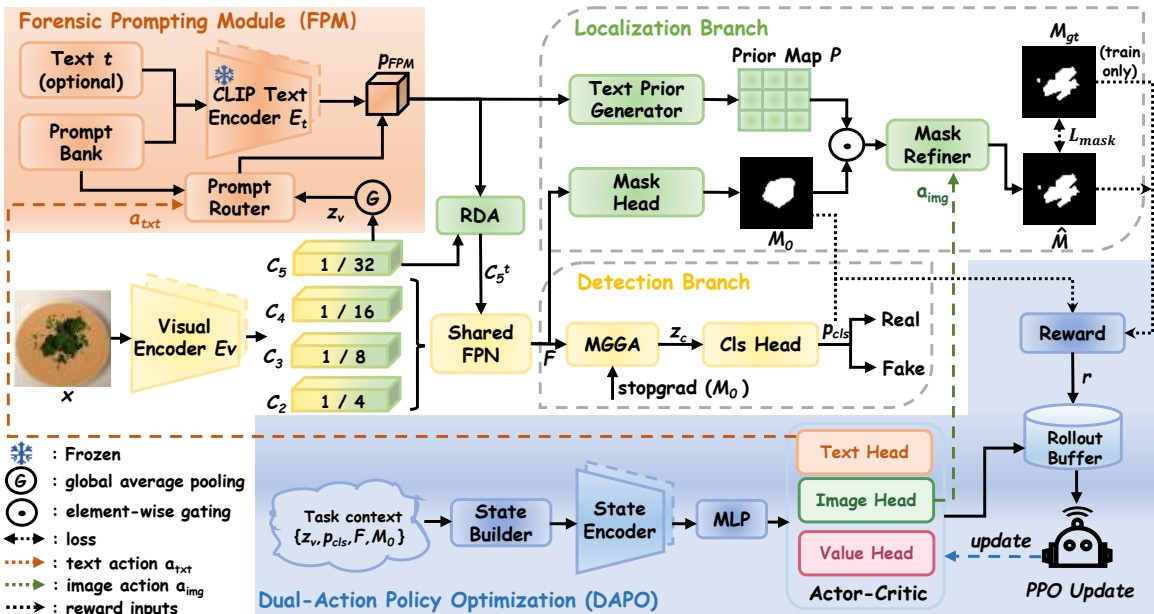

*Figure 1.* Overall framework of FPDA. FPM constructs a structured and reproducible forensic prompt bank as an evidence-centric conditioning space, and derives the final prompt representation with optional external text fused through a reliability gate. The prompt representation conditions a dual-branch network for forgery detection and localization. DAPO provides per-image dual-action control through two discrete decisions $(a_{\text{txt}}, a_{\text{img}})$, where $a_{\text{txt}}$ governs prompt routing and $a_{\text{img}}$ schedules mask refinement.

prompt routing and image-side refinement scheduling, to adapt evidence selection and localization refinement at the sample level. Overall, we make the following key contributions:

- We propose FPDA for joint forgery detection and localization, and formulate a unified text-optional input protocol that accommodates both text-missing and text-available settings under a single inference formulation.

- We develop FPM with a structured and reproducible forensic prompt bank and a sample-adaptive Prompt Router, providing stable evidence-centric conditioning while avoiding irreproducible open-ended prompting.

- We introduce DAPO as a dual-action mechanism for sample-adaptive evidence usage, where the text-side action routes forensic evidence and the image-side action schedules mask refinement, mitigating localization drift and improving spatial consistency.

## 2. Related Work

**Forgery Detection and Localization.** Classical image forensics exploits low-level acquisition and compression traces, including sensor noise patterns, demosaicking/interpolation artifacts, and JPEG quantization effects. Residual-based representations (e.g., SRM-style models) suppress semantic content and amplify weak forensic cues (Fridrich & Kodovsky, 2012). Early deep forensic models

further encourage content-agnostic evidence by imposing constrained convolutional layers that promote high-pass responses and reduce content leakage (Bayar & Stamm, 2016). As manipulations span heterogeneous operations (splicing, copy-move, inpainting, enhancement, and their combinations), modern end-to-end frameworks increasingly address detection and localization jointly, e.g., via anomalous-feature formulations for manipulation localization (Wu et al., 2019) and by synthesizing hard positive composites followed by coarse-to-fine segmentation and refinement (Zhou et al., 2020), while transformer-based designs model region-level consistency to support joint predictions (Wang et al., 2022). Nevertheless, forensic evidence can be non-stationary under cross-dataset shifts and post-processing, and localization quality remains sensitive to unreliable cues and fixed refinement strategies, motivating more robust and spatially reliable representations that target ubiquitous, manipulation-agnostic inconsistencies (Kong et al., 2025). In parallel, diffusion-era studies leverage reconstruction and denoising behavior as interpretable signals for detecting synthetic content and supporting localization, such as reconstruction-error-based detectors (Wang et al., 2023) and diffusion-prior-driven IFDL frameworks (Yu et al., 2024).

**Forensic Prompting.** Vision-language pretraining provides reusable text anchors for organizing visual evidence and enabling open-vocabulary transfer (Radford et al., 2021). Prompt learning has moved from handcrafted tem-

plates to parameter-efficient context optimization (e.g., CoOp/CoCoOp) and lightweight/test-time adaptation for distribution shift (Zhou et al., 2022b; Jia et al., 2022; Zhang et al., 2022; Gao et al., 2024; Shu et al., 2022; Khattak et al., 2023). For dense prediction, CLIP-derived supervision and promptable segmenters enable modular prompt-driven localization (Zhou et al., 2022a; Kirillov et al., 2023). In image forensics, however, descriptions are often missing or noisy, and unconstrained text can cause prompt-side semantic drift across detection and localization. This motivates structured and reproducible forensic prompting, where text serves as controllable, task-aligned evidence-level conditioning rather than an open-ended prompt source.

**Policy Optimization.** Policy optimization aims to learn a policy that performs sample-wise discrete control by maximizing the expected return. In practice, proximal policy optimization (PPO) is often adopted due to its robust and stable updates under noisy rewards (Williams, 1992; Schulman et al., 2015; 2017). Meanwhile, when the decision horizon collapses to a single-step choice with immediate feedback, contextual multi-armed bandits offer a standard abstraction for balancing exploration and exploitation (Lattimore & Szepesvári, 2020). Along similar lines, reinforcement learning has been used for prompting-related discrete search and composition, which naturally supports a policy-learning view of prompt selection and fusion (Deng et al., 2022). However, in joint forgery detection and localization, the challenge is not a single discrete knob but two coupled decisions: text-side prompt routing and image-side scheduling of localization refinement. Optimizing these decisions in isolation may overfit to biased conditioning signals, leading to spatial prediction shifts and evidence inconsistency across the two tasks (Kong et al., 2025; Yu et al., 2024). These observations motivate a feedback-conditioned dual-action formulation that determines which forensic evidence should be activated and how the selected evidence should guide localization refinement.

# 3. Methodology

## 3.1. Forensic Prompting Module (FPM)

Text conditioning can provide semantic priors for visual forensics, yet open-ended prompting is difficult to reproduce and constrain. Moreover, the optional text input $t$ may be missing, weakly aligned, or semantically noisy, which can introduce prompt-side perturbations and destabilize the conditioning used for forgery detection and localization. To address these issues, FPM constructs a structured and reproducible forensic prompt bank as an evidence-centric conditioning space, uses instance-adaptive routing to obtain a bank-based prompt representation, and incorporates optional text in a reliability-aware manner through gated fusion, as illustrated in Figure 2.

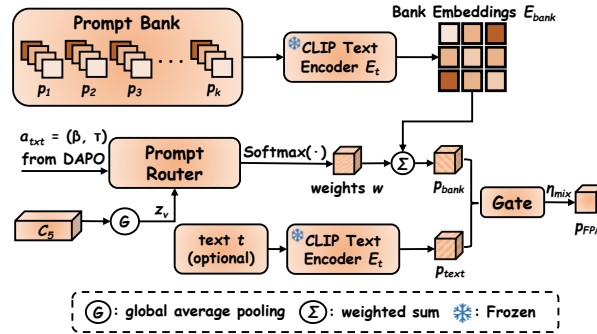

*Figure 2.* Forensic Prompting Module (FPM). A frozen CLIP text encoder $E_t$ encodes the prompt bank and optional text $t$ into $E_{\text{bank}}$ and $p_{\text{text}}$. Conditioned on the visual context $z_v$ and text action $a_{\text{txt}} = (\beta, \tau)$, the Prompt Router predicts weights $w$ to aggregate $E_{\text{bank}}$ into $p_{\text{bank}}$. The Gate estimates $\eta_{\text{mix}}$ and fuses $p_{\text{bank}}$ with $p_{\text{text}}$ to obtain $p_{\text{FPM}}$.

### 3.1.1. PROMPT BANK

FPM maintains a structured Prompt Bank $\mathcal{P} = \{p_k\}_{k=1}^K$ to define an evidence-centric conditioning space for visual forensics. The bank organizes observable forensic cues into a stable and routable evidence vocabulary, enabling the Prompt Router to perform sample-adaptive evidence selection.

Specifically, we organize evidence categories according to their forensic source, visual manifestation, and relevance to forgery detection and localization. The categories include generative traces, compression inconsistencies, resampling/interpolation artifacts, texture/statistical anomalies, boundary/blending irregularities, illumination/shadow inconsistencies, face-specific inconsistencies, text-rendering anomalies, and other contextual anomalies. These complementary categories provide controllable evidence priors for both image-level discrimination and pixel-level localization.

To ensure reproducibility, we adopt a template-guided construction. For each evidence category $c$, we use evidence-oriented phrasings such as "forensic evidence of $c$", "image region with $c$", and "possible manipulated area showing $c$". Each prompt is then encoded by a frozen CLIP text encoder $E_t$ to form bank embeddings:

$$E_{\text{bank}} = \big[ E_t(p_1); \ E_t(p_2); \ \cdots \ ; \ E_t(p_K) \big] \in \mathbb{R}^{K \times D}. \quad (1)$$

Here, $E_t$ provides a shared semantic space for the Prompt Bank and optional text. Freezing $E_t$ stabilizes the semantic anchors of the evidence vocabulary, reduces prompt-side drift, and provides a consistent basis for the subsequent Prompt Router and reliability-gated fusion.

### 3.1.2. PROMPT ROUTER

Since salient forensic cues vary across samples, FPM employs a Prompt Router to adaptively aggregate Prompt Bank

embeddings. Specifically, we extract the visual context $z_v$ from the top-level $1/32$ feature map $C_5$ produced by the visual encoder $E_v$ via global average pooling, and feed it to a lightweight router network $\phi_{\mathrm{rt}}(\cdot)$ to produce routing logits.

DAPO provides the text-side action $a_{\mathrm{txt}} = (\beta, \tau)$ to modulate the routing distribution, where $\beta$ denotes a routing bias for adjusting evidence preference and $\tau > 0$ denotes a temperature coefficient for controlling distribution sharpness. The routing weights are computed as:

$$w = \mathrm{Softmax}\left(\frac{\phi_{\mathrm{rt}}(z_v) + \beta}{\tau}\right) \in \mathbb{R}^K. \qquad (2)$$

Then, the Prompt Router aggregates Prompt Bank embeddings according to $w$ to obtain the bank-based prompt representation:

$$p_{\mathrm{bank}} = \sum_{k=1}^{K} w_k E_t(p_k) \in \mathbb{R}^D. \qquad (3)$$

Here, $p_{\mathrm{bank}}$ represents the sample-selected combination of forensic evidence and serves as the bank-derived prompt representation for the subsequent reliability-gated fusion.

### 3.1.3. RELIABILITY-GATED FUSION

The optional text $t$ is encoded into the same semantic space by the frozen encoder and normalized as $p_{\mathrm{text}} = \mathrm{Norm}(E_t(t))$, where $\mathrm{Norm}(\cdot)$ denotes $\ell_2$-normalization with a small $\epsilon$ for numerical stability. A gating function $\mathrm{Gate}(p_{\mathrm{bank}}, p_{\mathrm{text}})$ predicts a mixing coefficient $\eta_{\mathrm{mix}} \in (0, 1)$ to regulate the contribution of optional text. The final prompt representation is obtained by controlled fusion:

$$p_{\mathrm{FPM}} = \mathrm{Norm}\left((1 - \eta_{\mathrm{mix}})p_{\mathrm{bank}} + \eta_{\mathrm{mix}}p_{\mathrm{text}}\right) \in \mathbb{R}^D. \qquad (4)$$

When $t$ is missing or invalid, we set $\eta_{\mathrm{mix}} = 0$, yielding the deterministic bank-only condition $p_{\mathrm{FPM}} = p_{\mathrm{bank}}$.

### 3.2. Dual-Action Policy Optimization (DAPO)

DAPO learns a sample-level evidence usage policy. Since different forged images exhibit different key cues and coarse-mask error patterns, the model needs to decide both which forensic evidence to emphasize and how to refine the localization result. To this end, DAPO introduces two coordinated discrete actions: the text-side action $a_{\mathrm{txt}}$ modulates evidence routing in the Prompt Router, while the image-side action $a_{\mathrm{img}}$ controls the refinement strategy in the Mask Refiner. The policy uses detection and localization feedback to coordinate evidence selection and mask refinement.

### 3.2.1. FORMULATION

**Setup.** For each sample, DAPO builds a detached policy state from the current task context and samples a joint action

$a = (a_{\mathrm{txt}}, a_{\mathrm{img}})$ from the policy $\pi_\theta(a \mid s)$. The sampled action is then applied to prompt routing and mask refinement in the action-conditioned forward pass, producing the detection prediction and the refined mask $\hat{M}$ for task feedback. The policy is optimized by maximizing the expected reward:

$$\max_\theta \mathbb{E}_{s, a\sim\pi_\theta(\cdot|s)} [r(s, a)]. \qquad (5)$$

**State.** We construct the policy state from detached task-context signals:

$$s = \mathrm{stopgrad}\left(\mathcal{B}(\mathrm{ctx})\right), \quad \mathrm{ctx} = (z_v, p_{\mathrm{cls}}, F, M_0). \quad (6)$$

Here, $\mathrm{ctx}$ denotes the task context shown in Figure 1, where $z_v = \mathrm{GAP}(C_5)$ is the global visual context, $p_{\mathrm{cls}}$ is used as a detached detection-context signal, $F$ is the fused feature, and $M_0$ is the coarse mask. These signals summarize the current sample state and are used only as policy inputs, preventing policy gradients from backpropagating into the main detection-localization network.

**Actor-Critic.** DAPO uses an Actor-Critic architecture, where the Actor has a shared encoder with two categorical heads for $a_{\mathrm{txt}}$ and $a_{\mathrm{img}}$, and the Critic predicts a value baseline $V_\phi(s)$. The advantage is computed as:

$$A(s, a) = r(s, a) - V_\phi(s). \qquad (7)$$

We optimize the Actor with the clipped PPO objective:

$$J_{\mathrm{ppo}}(\theta) = \mathbb{E}\left[\min\left(\rho(\theta)A, \mathrm{clip}(\rho(\theta), 1 - \epsilon, 1 + \epsilon)A\right)\right], \qquad (8)$$

where $\rho(\theta) = \pi_\theta(a \mid s)/\pi_{\theta_{\mathrm{old}}}(a \mid s)$. The Actor is updated by maximizing $J_{\mathrm{ppo}}(\theta)$, while the Critic regresses $V_\phi(s)$ to the reward. The policy and value networks are updated using mini-batches of tuples $(s, a, r)$.

### 3.2.2. DUAL-ACTION DESIGN

DAPO decomposes sample-adaptive control into two coordinated discrete actions. The text-side action determines which forensic evidence should be emphasized by modulating prompt routing, while the image-side action determines how the coarse mask should be refined using the prior map, as summarized in Algorithm 1.

**Text action $a_{\mathrm{txt}}$.** The text-side action $a_{\mathrm{txt}} \in \mathcal{A}_{\mathrm{txt}}$ operates on the Prompt Router. It specifies a routing regime that modulates the distribution over the Prompt Bank, thereby changing how forensic cues are composed into the bank-based prompt representation $p_{\mathrm{bank}}$. Through the final prompt representation $p_{\mathrm{FPM}}$, this action controls the evidence emphasis used to condition both detection and localization branches.

**Image action $a_{\mathrm{img}}$.** The image-side action $a_{\mathrm{img}} \in \mathcal{A}_{\mathrm{img}}$ operates on the Mask Refiner. It specifies the refinement strategy used to update the coarse mask $M_0$ under the prior

**Algorithm 1** DAPO training for FPDA

**Require:** Training set $\mathcal{D}_f$; FPDA parameters $\omega$; policy/value $(\theta, \phi)$; PPO hyperparameters $(\epsilon, N_{\text{ppo}})$.
**Ensure:** Trained $(\omega, \theta, \phi)$.
1: Initialize $\omega, \theta, \phi$.
2: **while** not converged **do**
3:     Sample $\{(x_i, t_i, y_i, M_{\text{gt}}^{(i)})\}_{i=1}^{B} \subset \mathcal{D}_f$.
4:     **for** $i = 1$ to $B$ **do**
5:         State: $s_i \leftarrow \text{stopgrad}(\mathcal{B}(\text{ctx}_i))$.     ▷ **context**
6:         Action: $a_i \sim \pi_\theta(\cdot \mid s_i)$, $a_i = (a_{\text{txt}}, a_{\text{img}})$. ▷ **dual**
7:         Predict: $(p_i, \hat{M}_i) \leftarrow f_\omega(x_i, t_i; a_i)$.     ▷ **forward**
8:         Reward: $r_i \leftarrow r(p_i, \hat{M}_i; y_i, M_{\text{gt}}^{(i)})$.     ▷ **train**
9:     **end for**
10:   Update $\omega$ by $\mathcal{L}_{\text{sup}}$.     ▷ **FPDA**
11:   Update $(\theta, \phi)$ by PPO for $N_{\text{ppo}}$ epochs.     ▷ **policy**
12: **end while**

map $P$, as detailed in the localization branch below. We use three refinement choices:

$$\mathcal{A}_{\text{img}} = \{\text{skip}, \text{strong}, \text{edge}\}. \tag{9}$$

Here, skip preserves a conservative update close to $M_0$, strong promotes region-level completion and connectivity, and edge emphasizes boundary consistency for contour alignment.

### 3.2.3. REWARD

**Detection Reward.** Let $p_{\text{cls}}[y]$ denote the predicted probability assigned to the ground-truth label $y \in \{\text{Real}, \text{Fake}\}$. We define a bounded detection reward:

$$r_{\text{det}} = \text{clip}\left(2p_{\text{cls}}[y] - 1, \; -1, \; 1\right). \tag{10}$$

**Localization Reward.** For samples with pixel-level annotations, the localization reward measures the quality improvement of the refined mask over the coarse mask. We first define a soft-IoU quality metric:

$$Q(M, M_{\text{gt}}) = \frac{\langle M, M_{\text{gt}} \rangle}{\langle M, 1 \rangle + \langle M_{\text{gt}}, 1 \rangle - \langle M, M_{\text{gt}} \rangle + \epsilon}. \tag{11}$$

Based on this metric, we use the bounded improvement of the refined mask $\hat{M}$ over the coarse mask $M_0$ as the localization reward:

$$r_{\text{loc}} = \text{clip}\left(2\left[Q(\hat{M}, M_{\text{gt}}) - Q(M_0, M_{\text{gt}})\right], -1, \; 1\right). \tag{12}$$

**Overall Reward.** The overall policy reward is defined as:

$$r(s, a) = r_{\text{det}} + \lambda_{\text{loc}} r_{\text{loc}}. \tag{13}$$

Here, $\lambda_{\text{loc}}$ balances the reward scales. The detection reward reflects whether evidence routing benefits image-level discrimination, while the localization reward evaluates whether

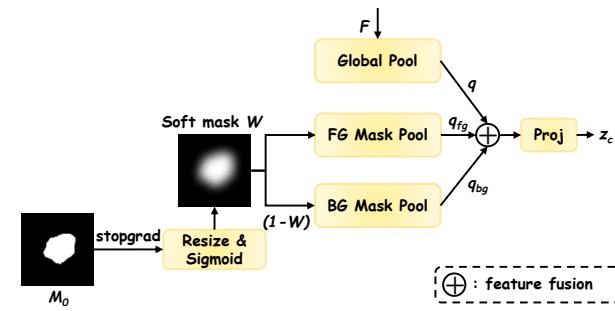

*Figure 3.* Mask-Guided Global Aggregation (MGGA). The stop-gradient coarse mask $M_0$ is converted to a soft mask $W$ for foreground/background pooling. MGGA combines $q$, $q_{\text{fg}}$, and $q_{\text{bg}}$ from $F$ to produce the classification feature $z_c$.

the selected refinement path improves the mask over $M_0$. Thus, the bounded reward provides task-level feedback for evidence selection and mask refinement, and serves as the policy signal for DAPO.

### 3.3. Dual-Branch Network

We adopt a dual-branch network to jointly perform image-level forgery detection and pixel-level forgery localization. Given an input image, the visual encoder produces multi-scale features $\{C_2, C_3, C_4, C_5\}$. Before multi-scale fusion, Residual Deformable Adaptation (RDA) injects the prompt representation $p_{\text{FPM}}$ into the top-level feature $C_5$ to obtain a prompt-modulated high-level feature (see Appendix A.2 for details). The modulated feature is then integrated with lower-level features by a SharedFPN to form a unified representation $F$, which serves as the shared input to both branches. Based on $F$, the localization branch predicts a coarse mask $M_0$ and further refines it, while the detection branch uses a stop-gradient mask cue for image-level classification. This design couples detection and localization on a shared evidence representation while avoiding destabilizing gradients from the detection objective to the mask prediction path.

#### 3.3.1. DETECTION BRANCH

The detection branch produces the image-level prediction $p_{\text{cls}}$. As illustrated in Figure 3, we apply Mask-Guided Global Aggregation (MGGA) on the shared feature $F$ to build a localization-aware classification representation. Specifically, the coarse mask $M_0$ is detached, resized, and normalized into a soft spatial weight $W = \sigma(\text{Resize}(\text{stopgrad}(M_0)))$. MGGA extracts a global descriptor $q$, a foreground descriptor $q_{\text{fg}}$ pooled with $W$, and a background descriptor $q_{\text{bg}}$ pooled with $1 - W$. The foreground descriptor focuses on suspicious regions, while the background descriptor provides complementary contrastive context for separating manipulated and authentic regions. These descriptors are then jointly fused into the classifica-

*Table 1.* Image-level forgery detection results on five datasets. AUC and balanced accuracy (Acc) are reported, where Acc is computed using a fixed threshold of 0.5. The best and second-best results are highlighted in bold and underlined, respectively.

| Methods | | CASIA v1 | | Columbia | | NIST16 | | COVERAGE | | CocoGlide | |
| --- | --- | --- | --- | --- | --- | --- | --- | --- | --- | --- | --- |
| | | AUC | Acc | AUC | Acc | AUC | Acc | AUC | Acc | AUC | Acc |
| ManTra-Net (Wu et al., 2019) | CVPR'19 | 0.817 | 0.520 | 0.810 | 0.500 | 0.795 | 0.500 | 0.849 | 0.801 | 0.778 | 0.500 |
| SPAN (Hu et al., 2020) | ECCV'20 | 0.797 | 0.600 | 0.998 | 0.950 | 0.804 | 0.597 | 0.937 | 0.590 | 0.475 | 0.491 |
| MVSS-Net (Chen et al., 2021) | ICCV'21 | 0.815 | 0.655 | 0.984 | 0.940 | 0.579 | 0.538 | 0.893 | 0.650 | 0.654 | 0.552 |
| PSCC-Net (Liu et al., 2022) | TCSVT'22 | 0.829 | 0.900 | 0.810 | 0.780 | 0.501 | 0.456 | 0.941 | 0.840 | 0.777 | 0.661 |
| IF-OSN (Wu et al., 2022) | CVPR'22 | 0.735 | 0.647 | 0.882 | 0.521 | 0.658 | 0.418 | 0.557 | 0.510 | 0.611 | 0.567 |
| CAT-Net (Kwon et al., 2022) | IJCV'22 | 0.942 | 0.843 | 0.977 | 0.805 | 0.750 | 0.500 | 0.680 | 0.635 | 0.667 | 0.586 |
| TruFor (Guillaro et al., 2023) | CVPR'23 | 0.833 | 0.811 | 0.996 | 0.983 | 0.760 | 0.662 | 0.871 | 0.685 | 0.752 | 0.639 |
| CoDE (Peng et al., 2024) | TIFS'24 | 0.769 | 0.551 | 0.803 | 0.507 | 0.685 | 0.401 | 0.577 | 0.505 | 0.702 | 0.505 |
| SAFIRE (Kwon et al., 2025) | AAAI'25 | 0.796 | 0.535 | 0.758 | 0.498 | 0.632 | 0.392 | 0.545 | 0.505 | 0.645 | 0.501 |
| FPDA (Ours) | | **0.975** | **0.936** | **0.999** | **0.997** | **0.902** | **0.875** | **0.983** | **0.965** | **0.941** | **0.892** |

tion feature:

$$z_c = \mathrm{Proj}\left([q;\ q_{\mathrm{fg}};\ q_{\mathrm{bg}}]\right), \qquad (14)$$

which is fed into the classification head to obtain $p_{\mathrm{cls}}$. Detaching $M_0$ allows the detection branch to exploit localization cues for spatially aware classification without back-propagating detection gradients into the mask prediction path.

### 3.3.2. LOCALIZATION BRANCH

The localization branch follows a coarse-to-refined pipeline. The Mask Head first predicts coarse mask logits from the shared feature $F$, which are upsampled and normalized to obtain the coarse mask:

$$M_0 = \sigma\left(\mathrm{Up}\left(\mathrm{MaskHead}(F)\right)\right). \qquad (15)$$

The Text Prior Generator predicts a prior map $P$ from $(F, p_{\mathrm{FPM}})$ to provide prompt-conditioned spatial guidance for refinement. The Mask Refiner takes $M_0$, $P$, and the image-side action $a_{\mathrm{img}}$ as inputs, where $a_{\mathrm{img}}$ selects the refinement strategy. The final refined mask is produced as:

$$\hat{M} = \sigma\left(\mathrm{Up}\left(\mathrm{MaskRefiner}\left(M_0,\ P,\ a_{\mathrm{img}}\right)\right)\right). \qquad (16)$$

Here, $\mathrm{Up}(\cdot)$ denotes upsampling to the input image resolution, and $\hat{M}$ is the final output for pixel-level forgery localization.

### 3.3.3. SUPERVISED OBJECTIVE

The dual-branch detection-localization network is trained with the joint supervised objective:

$$\mathcal{L}_{\mathrm{sup}} = \mathcal{L}_{\mathrm{det}}(p_{\mathrm{cls}}, y) + \lambda_{\mathrm{mask}}\mathcal{L}_{\mathrm{mask}}(\hat{M}, M_{\mathrm{gt}}). \qquad (17)$$

Here, $\mathcal{L}_{\mathrm{det}}$ supervises the image-level forgery prediction, and $\mathcal{L}_{\mathrm{mask}}$ supervises the final localization mask $\hat{M}$ with the ground-truth mask $M_{\mathrm{gt}}$. The coefficient $\lambda_{\mathrm{mask}}$ balances the two loss terms.

## 4. Experiments

### 4.1. Datasets and Evaluation Metrics

We evaluate FPDA on multiple public datasets covering diverse forgery mechanisms, grouped by manipulation paradigm: (i) pixel-level annotated manual manipulations, including CASIA v1 (Dong et al., 2013), COVERAGE (Wen et al., 2016), NIST16 (Guan et al., 2019), and Columbia (Ng et al., 2004); (ii) diffusion-era generative editing with CocoGlide (Guillaro et al., 2023); (iii) face forgery benchmark OpenForensics (Le et al., 2021); and (iv) SID_Set_description, a text-described subset derived from SIDA with masks and per-image descriptions (Huang et al., 2025), used for text-enabled FPM mechanism analysis. We report image-level detection under a binary Real/Forged setting, where Forged merges Full Synthetic and Tampered, and pixel-level localization on samples with valid ground-truth masks. For OpenForensics, we report localization-only results. For each dataset, we use a 7:1:2 train/validation/test split, where the training split is used for model learning, the validation split for model selection and early stopping, and the test split for final reporting.

For image-level detection, the main comparison reports AUC and balanced accuracy (Acc), where Acc is computed using a fixed threshold of 0.5. We additionally report detection F1 in ablation studies to reflect the precision–recall balance. For pixel-level localization, we report pixel-level F1 and IoU using a fixed mask binarization threshold of 0.5.

### 4.2. Results Analysis

**Detection Results.** Table 1 reports image-level detection results on five benchmarks. For baselines, we cite reported numbers when the evaluation protocol is directly comparable; otherwise, we re-evaluate them under the same evaluation protocol. FPDA obtains favorable AUC and balanced Acc across these datasets. On CocoGlide, FPDA achieves

*Table 2.* Pixel-level forgery localization results on five datasets. Pixel-level F1 and IoU are reported using a fixed mask binarization threshold of 0.5. The best and second-best results are highlighted in bold and underlined, respectively. "–" indicates that no directly comparable result is available.

| Methods | | CASIA v1 | | Columbia | | NIST16 | | OpenForensics | | CocoGlide | |
|---|---|---|---|---|---|---|---|---|---|---|---|
| | | F1 | IoU | F1 | IoU | F1 | IoU | F1 | IoU | F1 | IoU |
| ManTra-Net (Wu et al., 2019) | CVPR'19 | 0.187 | 0.111 | 0.452 | 0.301 | 0.160 | 0.098 | 0.551 | – | 0.516 | 0.441 |
| SPAN (Hu et al., 2020) | ECCV'20 | 0.143 | 0.112 | 0.562 | 0.390 | 0.211 | 0.156 | 0.089 | – | 0.296 | 0.221 |
| MVSS-Net (Chen et al., 2021) | ICCV'21 | 0.432 | 0.379 | 0.665 | 0.588 | 0.305 | 0.248 | 0.716 | 0.607 | 0.333 | 0.257 |
| CAT-Net (Kwon et al., 2022) | IJCV'22 | 0.581 | 0.509 | 0.584 | 0.469 | 0.269 | 0.196 | 0.788 | 0.684 | 0.363 | 0.288 |
| PSCC-Net (Liu et al., 2022) | TCSVT'22 | 0.335 | 0.232 | 0.503 | 0.360 | 0.173 | 0.108 | 0.575 | 0.449 | 0.421 | 0.332 |
| IF-OSN (Wu et al., 2022) | CVPR'22 | 0.509 | 0.465 | 0.706 | 0.607 | 0.268 | 0.199 | 0.784 | 0.698 | 0.264 | 0.207 |
| HiFi-Net (Guo et al., 2023) | CVPR'23 | 0.092 | 0.078 | 0.833 | 0.741 | 0.128 | 0.078 | – | – | 0.211 | 0.148 |
| TruFor (Guillaro et al., 2023) | CVPR'23 | 0.696 | 0.633 | 0.798 | 0.740 | 0.362 | 0.291 | 0.802 | – | 0.360 | 0.292 |
| CoDE (Peng et al., 2024) | TIFS'24 | 0.723 | 0.637 | 0.881 | 0.844 | 0.420 | 0.339 | 0.761 | 0.679 | 0.489 | 0.387 |
| SAFIRE (Kwon et al., 2025) | AAAI'25 | 0.299 | 0.238 | 0.900 | 0.884 | 0.168 | 0.138 | – | – | 0.479 | 0.404 |
| FakeShield (Xu et al., 2025) | ICLR'25 | 0.600 | 0.540 | 0.750 | 0.670 | 0.370 | 0.320 | – | – | 0.592 | 0.518 |
| PIM (Kong et al., 2025) | TPAMI'25 | 0.566 | 0.512 | 0.680 | 0.604 | 0.280 | 0.225 | – | – | – | – |
| FPDA (Ours) | | **0.738** | **0.652** | **0.904** | **0.890** | **0.686** | **0.580** | **0.842** | **0.759** | **0.749** | **0.692** |

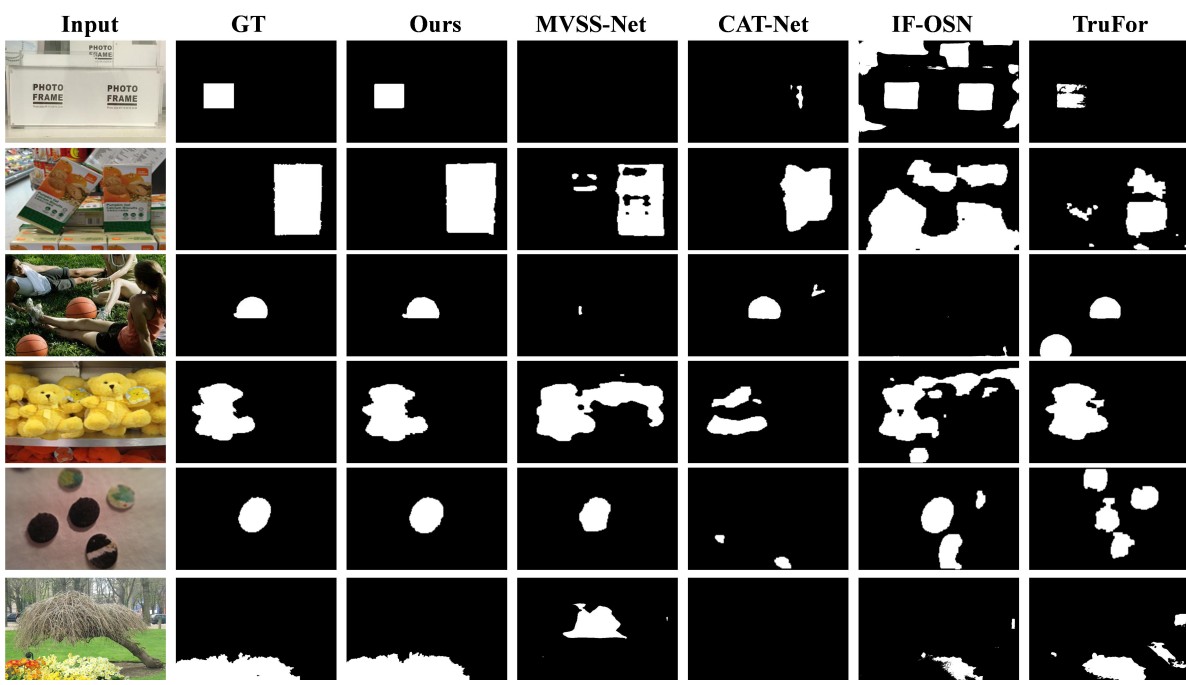

*Figure 4.* Qualitative localization comparison. Visual results on representative samples show the predicted masks of different methods. FPDA produces region responses that are more consistent with the ground-truth masks in these examples.

0.941 AUC and 0.892 Acc, suggesting that the model remains effective under diffusion-era generative editing. On NIST16 and COVERAGE, FPDA obtains 0.902/0.875 and 0.983/0.965 in AUC/Acc, respectively; on CASIA v1 and Columbia, it obtains 0.975/0.936 and 0.999/0.997. Overall, these results are consistent with our detection–localization collaboration design, where localization cues provide spatial references for image-level detection.

**Localization Results.** Table 2 reports pixel-level localization results on five benchmarks. FPDA performs favorably in terms of both F1 and IoU. On NIST16, FPDA obtains 0.686/0.580 (F1/IoU), compared with 0.420/0.339 from CoDE. On CocoGlide, it obtains 0.749/0.692, compared with 0.592/0.518 from FakeShield. These results are consistent with the coarse-to-refined localization design, where the coarse mask provides region-level cues and the refinement

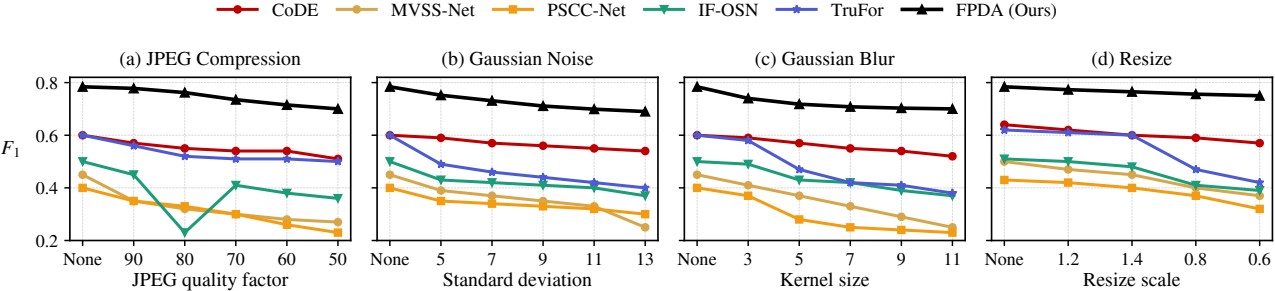

*Figure 5.* Robustness comparison under JPEG compression, Gaussian noise, Gaussian blur, and resizing. We report aggregated pixel-level $F_1$ for forgery localization over all test datasets; higher values indicate better performance.

*Table 3.* Component ablation on CocoGlide. Detection (AUC/F1) and localization (F1/IoU) are reported with a fixed mask threshold of 0.5. For +DAPO without FPM, the text-side action $a_{\text{txt}}$ is inactive, leaving only $a_{\text{img}}$ active.

| Setting | FPM | DAPO | Det. | | Loc. | |
|---|---|---|---|---|---|---|
| | | | AUC | F1 | F1 | IoU |
| Base | × | × | 0.905 | 0.835 | 0.690 | 0.620 |
| +FPM | ✓ | × | 0.925 | 0.860 | 0.715 | 0.650 |
| +DAPO | × | ✓ | 0.915 | 0.848 | 0.730 | 0.665 |
| Full | ✓ | ✓ | **0.941** | **0.901** | **0.749** | **0.692** |

*Table 4.* FPM ablation on SID_Set_description. Prompt Bank (B), Prompt Router (R), and optional text input (T) are progressively enabled. The last two rows compare optional text fusion without and with reliability gating. Detection (AUC/F1) and localization (F1/IoU) are reported.

| Variant | B | R | T | Det. | | Loc. | |
|---|---|---|---|---|---|---|---|
| | | | | AUC | F1 | F1 | IoU |
| w/o FPM | × | × | × | 0.985 | 0.970 | 0.662 | 0.571 |
| + Bank | ✓ | × | × | 0.989 | 0.975 | 0.670 | 0.580 |
| + Router | ✓ | ✓ | × | 0.993 | 0.979 | 0.679 | 0.588 |
| + Text (w/o gate) | ✓ | ✓ | ✓ | 0.992 | 0.978 | 0.676 | 0.586 |
| + Text (gated) | ✓ | ✓ | ✓ | **0.996** | **0.983** | **0.687** | **0.594** |

stage uses prompt-conditioned guidance for mask refinement.

**Qualitative Comparisons.** Figure 4 shows localization visualizations compared with MVSS-Net, CAT-Net, IF-OSN, and TruFor. On these representative samples, FPDA generally produces masks that are more consistent with the ground-truth regions, with fewer scattered responses in non-tampered areas.

**Robustness Evaluation.** To assess the impact of common post-processing on localization, we evaluate JPEG compression, Gaussian noise, Gaussian blur, and resizing, and report the aggregated pixel-level $F_1$ curves over all test datasets in Figure 5. At representative severe levels, FPDA maintains

*Table 5.* DAPO action ablation on NIST16. The text-side action $a_{\text{txt}}$ and image-side action $a_{\text{img}}$ are ablated. Detection (AUC/F1) and localization (F1/IoU) are reported.

| Variant | $a_{\text{txt}}$ | $a_{\text{img}}$ | Det. | | Loc. | |
|---|---|---|---|---|---|---|
| | | | AUC | F1 | F1 | IoU |
| None | × | × | 0.958 | 0.944 | 0.660 | 0.552 |
| Txt | ✓ | × | 0.964 | 0.950 | 0.668 | 0.560 |
| Img | × | ✓ | 0.961 | 0.947 | 0.674 | 0.569 |
| Txt+Img | ✓ | ✓ | **0.972** | **0.960** | **0.686** | **0.580** |

$F_1$ scores of 0.70, 0.69, 0.70, and 0.75 under JPEG quality 50, Gaussian noise with standard deviation 13, Gaussian blur with kernel size 11, and resize scale 0.6, respectively. Compared with CoDE and TruFor, FPDA shows smaller performance degradation under these settings, which is consistent with the observed degradation trends.

### 4.3. Ablation Studies

**Component Ablation.** As shown in Table 3, we evaluate the contributions of FPM and DAPO on CocoGlide under the same protocol. Compared with the base model, adding FPM improves detection AUC/F1 from 0.905/0.835 to 0.925/0.860, while also improving localization. Adding DAPO increases localization F1/IoU from 0.690/0.620 to 0.730/0.665, suggesting that image-side refinement control contributes to localization refinement. Since FPM is disabled in this setting, the text-side action is inactive and the gain mainly comes from the image-side action. The full model further reaches 0.941/0.901 in detection and 0.749/0.692 in localization, suggesting that structured prompting and dual-action control are complementary.

**Ablation on FPM.** Table 4 studies the internal design of FPM on SID_Set_description by progressively enabling the Prompt Bank (B), Prompt Router (R), and optional text fusion (T). Introducing the Prompt Bank improves detection AUC/F1 from 0.985/0.970 to 0.989/0.975, and adding the Prompt Router further improves the results to 0.993/0.979

*Table 6.* Reward ablation on CASIA v1. The detection reward $r_{\text{det}}$ and localization reward $r_{\text{loc}}$ are ablated. Detection (AUC/F1) and localization (F1/IoU) are reported.

| Variant | $r_{\text{det}}$ | $r_{\text{loc}}$ | Det. | | Loc. | |
|---|---|---|---|---|---|---|
| | | | AUC | F1 | F1 | IoU |
| w/o $r_{\text{det}}$ | × | ✓ | 0.953 | 0.915 | 0.732 | 0.642 |
| w/o $r_{\text{loc}}$ | ✓ | × | 0.972 | 0.946 | 0.702 | 0.620 |
| Full reward | ✓ | ✓ | **0.975** | **0.951** | **0.738** | **0.652** |

and 0.679/0.588 in localization F1/IoU, showing the benefit of structured evidence anchors and sample-adaptive routing. Directly using optional text without reliability gating slightly degrades performance compared with +Router, suggesting that weakly aligned descriptions may disturb bank-based conditioning. With reliability gating, the text-enabled variant reaches the best results, 0.996/0.983 in detection and 0.687/0.594 in localization, indicating that reliable text can serve as a complementary cue.

**Ablation on Dual-Action Control.** We conduct an action-set ablation of DAPO on NIST16, as summarized in Table 5. Compared with no action control, enabling the text-side action improves detection AUC/F1 from 0.958/0.944 to 0.964/0.950, suggesting its role in prompt routing and evidence selection. Enabling the image-side action yields larger gains in localization, improving F1/IoU from 0.660/0.552 to 0.674/0.569, which supports its role in refinement scheduling. Jointly using both actions further improves the results to 0.972/0.960 in detection and 0.686/0.580 in localization, suggesting that text-side evidence selection and image-side refinement scheduling provide complementary benefits.

**Reward Ablation.** We ablate the reward design of DAPO on CASIA v1 by removing either the detection reward $r_{\text{det}}$ or the localization reward $r_{\text{loc}}$, as shown in Table 6. Removing $r_{\text{det}}$ reduces detection AUC/F1 from 0.975/0.951 to 0.953/0.915, while removing $r_{\text{loc}}$ mainly affects localization, decreasing F1/IoU from 0.738/0.652 to 0.702/0.620. Using both reward terms gives the best results across detection and localization, suggesting that the two rewards provide complementary task feedback for DAPO.

## 5. Conclusion

We presented FPDA, a vision-language framework for joint image forgery detection and localization with optional text input. FPDA constructs a structured and reproducible forensic evidence space through FPM, enabling stable conditioning when external descriptions are missing, noisy, or weakly aligned. Built on this evidence space, DAPO learns sample-level dual-action control: the text-side action determines which forensic evidence should be emphasized, while the image-side action determines how the selected evidence

should guide mask refinement. Experiments across conventional manipulations, diffusion-era edits, and common post-processing perturbations suggest that structured evidence modeling and adaptive evidence usage improve both detection and localization under comparable evaluation protocols. Future work will explore richer forensic evidence spaces and more general action designs for robust forgery localization under complex real-world processing chains.

## Acknowledgements

This work was supported in part by the National Natural Science Foundation of China under Grant U24B20179, in part by the Natural Science Foundation of Hebei Province under Grant F2024202017, and in part by the Central Guiding Local Technology Development Fund Project under Grant 246Z0106G.

## Impact Statement

This work aims to improve the reliability of image forensics by enhancing forgery detection and localization, which can support content authentication, misinformation mitigation, and digital integrity analysis. As with other forensic technologies, it may also be misused or motivate adversaries to develop more evasive manipulation methods. We mitigate these potential risks by focusing on detection and localization, transparently reporting evaluation protocols, and encouraging responsible deployment with appropriate oversight.

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

# A. Appendix

## A.1. Experiments Implementation Details

All experiments are implemented in PyTorch and run on two NVIDIA A30 GPUs. Inputs are resized to $512 \times 512$ by default. We optimize the model with AdamW using a base learning rate of $1 \times 10^{-4}$ and weight decay of $1 \times 10^{-4}$. We apply linear warm-up for the first 2,000 updates, followed by cosine annealing with a minimum learning-rate ratio of 0.1. The model is trained for 20,000 updates with a total batch size of 8. We validate and save checkpoints every 500 updates, and report EMA results with a decay of 0.999. Unless otherwise specified, mixed precision is disabled, and the random seed is fixed to 42.

## A.2. Residual Deformable Adaptation (RDA)

RDA adapts the top-level feature $C_5$ through a prompt-conditioned deformable-attention block, where the final prompt representation $p_{\text{FPM}}$ controls the sampling offsets (Figure 6). Specifically, we compute $u_p = \text{MLP}(p_{\text{FPM}})$ and $\tilde{C}_5 = \text{Conv}_{1 \times 1}(C_5)$, and obtain the attended feature $U = \text{DefAttn}(\tilde{C}_5, \Delta(u_p))$, where $\Delta(\cdot)$ maps the prompt code to offset parameters. RDA then applies a residual update:

$$C_5^t = C_5 + \text{Conv}_{1 \times 1}(U). \tag{18}$$

Applying RDA only at $C_5$ provides a stable prompt-conditioned control point before multi-scale fusion while keeping the overhead small. Compared with channel-wise modulation, prompt-conditioned deformable sampling enables spatially selective adaptation, which better matches local and irregular forensic traces.

## A.3. Prompt Routing Heatmap

We visualize the evidence-aware routing behavior of FPM in Figure 7. For each test sample, we record the router weights $w$ in Eq. (2), aggregate them over the evidence categories $C$ in the structured prompt bank, and average the routing mass within each dataset/operation subset. Each row is normalized to sum to 1. The heatmap shows operation-dependent

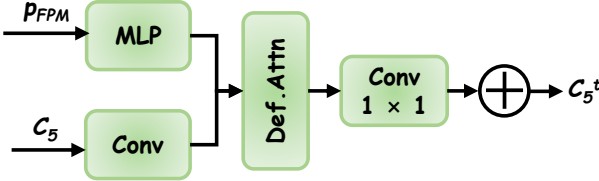

Figure 6. RDA module. The prompt representation $p_{\text{FPM}}$ is encoded to generate deformable-attention offsets, which adapt the top-level feature $C_5$. The attended feature is projected by a $1 \times 1$ convolution and added back residually to produce the prompt-modulated feature $C_5^t$.

Table 7. Computational overhead on Columbia. All variants use the same input resolution, inference setting, and device. Latency is reported in milliseconds, and memory denotes peak GPU memory.

| Variant | Params(M) | FLOPs(G) | Lat.(ms) | Mem.(GB) |
|---|---|---|---|---|
| Visual-only | 46.2 | 158.0 | 27.8 | 8.1 |
| + FPM | 46.9 | 159.2 | 28.4 | 8.2 |
| + DAPO | 47.1 | 159.5 | 28.8 | 8.3 |
| FPDA | 47.8 | 160.0 | 29.2 | 8.4 |

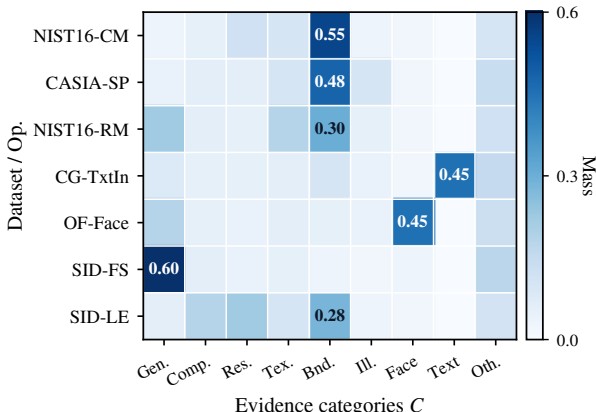

Figure 7. Prompt routing heatmap. Each cell shows the row-normalized routing mass assigned to an evidence category, obtained by aggregating router weights $w$ in Eq. (2) over each dataset/operation subset. Darker colors indicate larger routing mass.

routing patterns, indicating that the Prompt Router performs sample-adaptive evidence selection rather than uniformly using the prompt bank.

## A.4. Compute Cost

We report the computational overhead of different variants on Columbia in Table 7, using the same input resolution, inference setting, and device. Compared with the visual-only baseline, FPDA adds only 1.6M parameters, 2.0G FLOPs, 1.4 ms latency, and 0.3 GB peak memory. The additional cost mainly comes from lightweight routing, gating, and policy/value heads. Since the text encoder is frozen and the prompt-bank embeddings can be precomputed, FPM does not require repeated text encoding during inference. DAPO controls existing routing and refinement modules without repeated forward evaluation.

## A.5. Cross-Domain Generalization

To further examine transferability to unseen generative-editing forgeries, we conduct an additional cross-domain experiment. Models are trained only on manual-tampering datasets, including CASIA v1, COVERAGE, NIST16, and

*Table 8.* Cross-domain generalization to unseen generative-editing forgeries. Models are trained on CASIA v1, COVERAGE, NIST16, and Columbia, and directly tested on CocoGlide without target-domain fine-tuning. We use the bank-only setting without external text.

| Method | Det. | | Loc. | |
|---|---|---|---|---|
| | AUC | F1 | F1 | IoU |
| TruFor (Guillaro et al., 2023) | 0.846 | 0.801 | 0.592 | 0.517 |
| CoDE (Peng et al., 2024) | 0.861 | 0.816 | 0.618 | 0.544 |
| FPDA (Ours) | **0.913** | **0.867** | **0.701** | **0.626** |

*Table 9.* Adaptive selector comparison on NIST16. All variants use the same state, action, and reward design. Detection (AUC/F1) and localization (F1/IoU) are reported.

| Selection Strategy | Det. | | Loc. | |
|---|---|---|---|---|
| | AUC | F1 | F1 | IoU |
| Non-adaptive | 0.958 | 0.944 | 0.660 | 0.552 |
| Soft selector | 0.965 | 0.951 | 0.671 | 0.561 |
| G-ST | 0.968 | 0.954 | 0.678 | 0.569 |
| REINFORCE | 0.970 | 0.957 | 0.682 | 0.575 |
| PPO | **0.972** | **0.960** | **0.686** | **0.580** |

Columbia, and are directly tested on CocoGlide without target-domain fine-tuning. We use the bank-only setting without external text to isolate the effect of the structured forensic prompt bank and sample-adaptive evidence usage. As shown in Table 8, FPDA achieves 0.913/0.867 in detection AUC/F1 and 0.701/0.626 in localization F1/IoU, showing better transfer results than the compared methods under this setting. This experiment is complementary to the per-dataset train/test protocol used in the main comparison.

### A.6. Adaptive Selector Comparison

To further analyze the role of adaptive action selection in DAPO, we compare different selectors under the same state, action, and reward design. As shown in Table 9, the non-adaptive variant obtains the lowest performance, while soft selection and Gumbel-Softmax already improve both detection and localization metrics. This suggests that sample-adaptive dual-action selection is more effective than a fixed evidence-usage strategy. Among the tested optimization choices, PPO performs best, reaching 0.972/0.960 in detection and 0.686/0.580 in localization. These results support the dual-action control formulation, where PPO serves as a practical optimizer for the coupled discrete decisions.

