# OpenReview forum: "Forensic Prompting with Dual-Action Policy Optimization for Vision-Language Forgery Detection and Localization"
_ICML.cc/2026/Conference — ICML 2026 regular_

### Official Review · Reviewer_G6k7 · 2026-03-07

**Soundness:** 2
**Presentation:** 2
**Significance:** 3
**Originality:** 2
**Overall Recommendation:** 3
**Confidence:** 4

**Summary:**

This paper proposes FPDA, a framework for joint forgery detection and localization that integrates vision-language prompting with policy optimization. The method consists of two main components: a Forensic Prompting Module (FPM) and a Dual-Action Policy Optimization (DAPO) module. Experiments across multiple benchmarks demonstrate improvements in detection AUC and localization IoU compared to existing methods.

**Compliance With Llm Reviewing Policy:**

Affirmed.

**Final Justification:**

While the authors' response clarifies some design choices, I remain concerned that the work primarily integrates established techniques,  leading to marginal novelty. The core mechanisms lack sufficient analytical justification. Consequently, I choose to maintain my original score. I appreciate the authors’ effort and encourage them to strengthen the paper in future revisions.

**Key Questions For Authors:**

See weaknesses.

**Limitations:**

yes

**Strengths And Weaknesses:**

Strengths:
1. This paper clearly points out the practical pain points such as the difficulty in reproducing open-ended prompting and the instability due to missing text or noise.
2. This paper evaluates performance across diverse forgery paradigms and includes robustness tests against common post-processing distortions.


Weaknesses:
1.  While the integration of structured prompting with policy-based control is well-executed, the core technical contributions are incremental extensions of existing paradigms. The paper would benefit from deeper justification for why this specific combination yields gains over simpler baselines.
2. DAPO introduces actor-critic, reward shaping, and rollout buffers. The paper does not sufficiently justify whether this complexity is necessary compared to simpler heuristic or attention-based adaptive mechanisms, especially given the marginal gains in some ablation settings.
3. This paper cites SIDA but does not include direct quantitative comparisons with this large multimodal model approach.
3. Table 4 shows that enabling text input without adding a gate actually slightly reduces performance, while adding a gate only slightly improves it. In real-world scenarios where text is missing, is the marginal benefit of multimodal design worth the extra complexity?

---

> ### Author Rebuttal · Authors · 2026-03-30
>
> We thank the reviewer for the feedback.
>
> **1. About the integration of structured prompting and policy-based control**
>
> FPM and DAPO address different bottlenecks rather than being simply stacked. FPM stabilizes **what evidence is represented** by constraining prompts to a deterministic, finite, evidence-centric space, reducing prompt-side drift from open-ended prompting and noisy/missing text. DAPO adaptively controls **how evidence is used** for each sample through prompt routing and refinement scheduling. Thus, the gain comes from coupling stable evidence representation with sample-adaptive evidence allocation, not from adding modules.
>
> This is consistent with our ablations. On CocoGlide, Base, +FPM, +DAPO, and Full achieve **0.905/0.835/0.690/0.620**, **0.925/0.860/0.715/0.650**, **0.915/0.848/0.730/0.665**, and **0.941/0.901/0.749/0.692** (Det. AUC / Det. F1 / Loc. F1 / Loc. IoU). FPM and DAPO improve different bottlenecks, and their combination performs best. This indicates that FPM mainly improves evidence representation/routing, while the text-side action contributes more to detection and the image-side action more to localization.
>
> **2. About the necessity of DAPO and PPO**
>
> Our aim is not RL for its own sake, but a **dual-action discrete control** formulation for joint detection/localization: the text-side action controls prompt routing, the image-side action controls refinement-path selection, and both are optimized jointly under direct task feedback. The challenge is coordinating two coupled discrete actions, rather than handling a long decision horizon.
>
> Under this formulation, PPO serves as a practical optimizer for coupled discrete control under joint task feedback. More precisely, **DAPO contributes the dual-action control formulation, while PPO provides practical optimization for it.**
>
> To address why simpler heuristics or non-RL mechanisms are not used, we compare selectors/optimizers under the **same state, action, and reward** design:
>
> | Selector / Optimizer | Det. AUC | Det. F1 | Loc. F1 | Loc. IoU |
> |---|---:|---:|---:|---:|
> | Non-adaptive | 0.958 | 0.944 | 0.660 | 0.552 |
> | Soft selector (non-RL) | 0.965 | 0.951 | 0.671 | 0.561 |
> | G-ST (non-RL) | 0.968 | 0.954 | 0.678 | 0.569 |
> | REINFORCE (RL) | 0.970 | 0.957 | 0.682 | 0.575 |
> | PPO (RL) | **0.972** | **0.960** | **0.686** | **0.580** |
>
> The gain first comes from **adaptive dual-action selection itself**, since even non-RL selectors improve over the non-adaptive baseline. Under the same state/action/reward design, PPO gives the best result among the tested optimizers. We therefore do not claim PPO is the only viable choice; under the current formulation, it performs best among those tested.
>
> **3. About SIDA quantitative comparison**
>
> A direct quantitative comparison with SIDA is meaningful only under a matched protocol. Our text-related experiment does not use the full SID-Set / SIDA benchmark, but **SID Set description**, a derived subset with per-image descriptions. In this paper, it is used for **mechanism analysis of FPM in a text-enabled setting**, not for framework-level benchmark comparison.
>
> For this reason, SIDA is not included in the main table: the dataset scope, text setting, and evaluation purpose are not directly aligned. Including it there would mix mechanism analysis with benchmark comparison and weaken comparability. We cite SIDA because SID Set description is derived from it, but the main table is restricted to protocol-comparable results.
>
> **4. About the value of the text-optional design when text is often missing**
>
> The text-optional protocol is not intended to make FPDA strongly depend on text for average gains. Its purpose is to support both **text-available** and **text-missing** cases within one formulation: when reliable text is available, it provides complementary cues; when text is missing or invalid, the model falls back to stable bank-only conditioning. Sec. 3.1.3 explicitly states that missing/invalid text degenerates to deterministic bank-only conditioning. Thus, frequent text absence in practice does not remove FPDA’s core capability; text is complementary rather than indispensable.
>
> The FPM ablation supports this: w/o FPM, +Bank, +Router, +Text (w/o gate), and +Text (gated) achieve **0.985/0.970/0.662/0.571**, **0.989/0.975/0.670/0.580**, **0.993/0.979/0.679/0.588**, **0.992/0.978/0.676/0.586**, and **0.996/0.983/0.687/0.594**, respectively. The main stable gain comes from **Prompt Bank + Router**; ungated text slightly hurts, while gated text performs best. This shows that text is useful when reliable, but should not disturb stable bank-only conditioning when noisy or absent.
>
> The design is also lightweight: compared with the visual-only baseline, full FPDA adds only **1.6M** parameters, **2.0G** FLOPs, **1.4 ms** latency, and **0.3 GB** peak memory. Hence, the added complexity is limited while enabling a unified protocol robust to missing text and able to exploit reliable text when available.

---

> > ### Author Rebuttal · Reviewer_G6k7 · 2026-04-02
> >
> > Thanks for the detailed response, which have clarified several aspects of the design rationale. However, I remain concerned that the core contribution still relies heavily on integrating established components without providing deeper analytical justification. Therefore, I maintain my original rating. I appreciate the authors' efforts and hope they can strengthen the manuscript.

---

> > > ### Author Response · Authors · 2026-04-03
> > >
> > > Thank you for your further feedback. We greatly appreciate your concern that the core contribution may still appear to rely heavily on the integration of existing components, without sufficiently deep analytical justification. In response to this point, we would like to further clarify the internal logic of our method from the perspective of the **dual-action mechanism for sample-adaptive evidence usage**.
> > >
> > > More specifically, the key question addressed in our work is: for each input sample, **which forensic evidence should be activated, and how should that evidence influence localization and refinement on the image side**. These two questions correspond to the two actions in the dual-action mechanism:
> > >
> > > - **text-side action**: routes and invokes forensic evidence in the prompt space, i.e., it determines *what evidence should be used for the current sample*;
> > > - **image-side action**: couples that evidence with image-side refinement, i.e., it determines *how the selected evidence should take effect*.
> > >
> > > These two actions correspond to two complementary aspects of evidence usage: **evidence selection** and **mode of evidence application**. The former determines which forensic semantics the model should attend to, while the latter determines how those semantics enter the local visual representation and localization process. Together, they form a complete sample-level **evidence-to-localization path**.
> > >
> > > From this perspective, the dual-action mechanism also reflects the higher-level structural relation in our method. The structured Prompt Bank defines a stable and reproducible forensic evidence space, while the dual-action mechanism further learns, under final detection and localization feedback, how to control that evidence space on a per-sample basis. Accordingly, the core of our method can be more clearly understood as a closed-loop formulation: **first constructing a controllable forensic evidence space, and then learning feedback-conditioned control over that space**. Within this formulation, the dual-action mechanism plays the key role at the control level.
> > >
> > > This interpretation is also consistent with our ablation results. To further illustrate this point, we compared static evidence usage, non-RL adaptive selection, and the jointly controlled mechanism optimized by PPO under the same state, action, and reward setting:
> > >
> > > | Selector / Optimizer | Det. AUC | Det. F1 | Loc. F1 | Loc. IoU |
> > > |---|---:|---:|---:|---:|
> > > | Non-adaptive | 0.958 | 0.944 | 0.660 | 0.552 |
> > > | Soft (non-RL) | 0.965 | 0.951 | 0.671 | 0.561 |
> > > | G-ST (non-RL) | 0.968 | 0.954 | 0.678 | 0.569 |
> > > | REINF (RL) | 0.970 | 0.957 | 0.682 | 0.575 |
> > > | **PPO (RL)** | **0.972** | **0.960** | **0.686** | **0.580** |
> > >
> > > This trend indicates that detection and localization performance improves consistently when evidence usage shifts from a static regime to a sample-adaptive one, and achieves the best results under the joint control setting. We therefore interpret the key performance gain as coming from the **feedback-driven usage mechanism over a structured evidence space**, rather than from any single optimizer alone.
> > >
> > > Accordingly, we would like to emphasize that the core idea of the paper is not an empirical integration of established components, but a unified design centered on **forensic evidence-space construction** and **feedback-conditioned control over that space**. The dual-action mechanism is the concrete realization of this unified design at the evidence-usage level.
> > >
> > > Finally, we sincerely thank you again for your detailed and insightful comments throughout the review process. Your feedback has helped us more clearly define and articulate the core technical contribution of the paper, and we will make this contribution statement more explicit in the camera-ready version.

---

### Official Review · Reviewer_4ttY · 2026-03-11

**Soundness:** 3
**Presentation:** 2
**Significance:** 3
**Originality:** 3
**Overall Recommendation:** 3
**Confidence:** 4

**Summary:**

FPDA tackles joint image forgery detection and localization by introducing a Forensic Prompting Module (FPM) that maintains a structured, CLIP-encoded prompt bank with reliability-gated fusion, providing stable and reproducible text conditioning regardless of whether external descriptions are available. A Dual-Action Policy Optimization (DAPO) module uses single-step PPO to make two per-image discrete decisions—selecting a prompt routing regime and a mask refinement path (skip/strong/edge)—to adaptively match conditioning and localization strategies to each sample's characteristics. Experiments across conventional manipulation, diffusion-based editing, and face forgery benchmarks show consistent improvements in both detection AUC and localization IoU over prior methods, with moderate robustness to common post-processing degradations.

**Compliance With Llm Reviewing Policy:**

Affirmed.

**Final Justification:**

While the authors' rebuttal has addressed some of my concerns, I maintain that the novelty and insight of this work remain limited, as several of the proposed methods have already been extensively explored in both AIGI detection and deepfake detection. Moreover, the underlying mechanisms of the core approach have not been sufficiently analyzed. For these reasons, I choose to retain my original score. That said, I sincerely appreciate the authors' effort and recognize the potential of this paper. I would encourage the authors to conduct a deeper analysis of domain-specific insights and integrate them more explicitly with their proposed method.

**Key Questions For Authors:**

- What is the necessity of using PPO-based reinforcement learning for DAPO? Since the problem is essentially a single-step contextual bandit, could simpler alternatives (e.g., Gumbel-Softmax, straight-through estimators) achieve comparable results? An ablation comparing RL against non-RL selection mechanisms would be helpful.
- What are the specific prompt texts in the Prompt Bank? How were the evidence categories and template families designed, and what criteria guided these choices? The current paper does not disclose the actual prompts, making it difficult to assess or reproduce the method.
- What is the exact training data configuration? Which datasets are used for training vs. testing? Are models trained separately per dataset or under a unified cross-dataset protocol? This information is critical for evaluating generalization claims.
- For the image action space Aimg={skip, strong, edge}\mathcal{A}_{\text{img}} = \{\text{skip, strong, edge}\}
Aimg​={skip, strong, edge}, how was this specific set of refinement paths determined? Have other action designs been explored, and how sensitive is the performance to the granularity and composition of the action space?

**Limitations:**

Yes

**Strengths And Weaknesses:**

Strengths:

- Using reinforcement learning to automatically select prompt routing regimes and mask refinement paths on a per-sample basis is a novel and interesting idea for image forensics.
- FPDA achieves consistently strong performance across multiple benchmarks covering diverse manipulation types, with notable margins over prior methods.
- The added computational overhead is minimal (only ~1.4ms latency and ~1.6M extra parameters), keeping the approach practical for real-world deployment.

Weaknesses:

- The overall framework is overly complex, and Figure 1 is visually overwhelming with numerous interconnected modules (FPM, Prompt Router, RDA, MGGA, Mask Refiner, DAPO with Actor-Critic, etc.), making the method difficult to follow. The necessity of certain design choices is questionable — for instance, is RDA (Residual Deformable Adaptation) truly necessary for injecting the prompt into C5C_5
C5​, as opposed to simpler conditioning mechanisms such as feature modulation or channel-wise scaling? Similarly, the MGGA module introduces three separate pooling streams (global/foreground/background) guided by a stop-gradient coarse mask, but the marginal benefit over simpler mask-guided pooling is not clearly justified.

- The necessity of using reinforcement learning is not convincingly established. DAPO is formulated as a single-step decision problem, which is essentially a contextual bandit — a much simpler abstraction that does not require the full PPO machinery (Actor-Critic, clipped surrogate objective, rollout buffer, etc.). The paper lacks comparisons against simpler discrete selection mechanisms, such as Gumbel-Softmax relaxation, straight-through estimators, or learned soft attention over the action space, which could achieve similar sample-adaptive behavior with significantly less complexity. Without such baselines, it is unclear whether the performance gains stem from the adaptive selection itself or from the specific use of RL.

- The prompt bank is encoded by a frozen CLIP text encoder, but the prompts in the bank are domain-specific forensic descriptions (e.g., compression inconsistencies, resampling artifacts) that are unlikely to be well-represented in CLIP's pretraining distribution. It is questionable whether the frozen encoder can produce semantically meaningful and discriminative embeddings for such specialized forensic vocabulary. Moreover, the actual prompt texts are never disclosed in the paper, making it impossible to assess the quality of the prompt design or reproduce the results — which is ironic given the paper's repeated emphasis on "reproducibility."

- The fairness of comparisons in Tables 1 and 2 is concerning. The authors state that baseline numbers are either cited from original papers or re-evaluated, but do not specify which baselines fall into which category. Since different papers may use different data splits (especially for datasets like CASIA v1 that lack an official train/test partition), preprocessing pipelines, and evaluation thresholds, mixing reported and re-evaluated numbers can introduce systematic biases. A fully controlled re-evaluation under a unified protocol would be more convincing.

---

> ### Author Rebuttal · Authors · 2026-03-30
>
> We thank the reviewer for the feedback.
>
> **1. About framework complexity, RDA, and MGGA**
>
> We agree Figure 1 is dense and will simplify it. The gains do not come from added complexity alone.
>
> **RDA.** RDA injects $p_{\mathrm{FPM}}$ once into $C_5$ before multi-scale fusion, enabling spatially selective high-level conditioning. Compared with channel-wise scaling / FiLM, it better matches the local, sparse, irregular evidence common in image forensics.
>
> | $C_5$ cond. | Det. AUC | Det. F1 | Loc. F1 | Loc. IoU |
> |---|---:|---:|---:|---:|
> | Ch.-wise | 0.932 | 0.887 | 0.731 | 0.672 |
> | FiLM | 0.935 | 0.891 | 0.737 | 0.679 |
> | RDA | **0.942** | **0.917** | **0.749** | **0.692** |
>
> **MGGA.** MGGA is a mask-guided, localization-safe classifier aggregator: G preserves context, FG focuses suspicious regions, B adds contrast, and sg blocks detection gradients through $M_0$ into localization.
>
> | Agg. | Det. AUC | Det. F1 | Loc. F1 | Loc. IoU |
> |---|---:|---:|---:|---:|
> | FG | 0.966 | 0.951 | 0.670 | 0.562 |
> | G+FG | 0.969 | 0.955 | 0.676 | 0.569 |
> | G+FG+B (no sg) | 0.968 | 0.954 | 0.671 | 0.564 |
> | MGGA | **0.972** | **0.961** | **0.686** | **0.581** |
>
> These results suggest that RDA and MGGA help because their designs better match forgery evidence and task coupling, not simply because the framework is more complex.
>
> **2. About PPO-based DAPO**
>
> We agree the manuscript should distinguish DAPO’s methodological contribution from PPO’s optimization role more clearly. DAPO introduces unified dual-action control, not RL for its own sake: $a_{\text{txt}}$ controls prompt routing, $a_{\text{img}}$ controls refinement-path selection, and both are optimized jointly to reduce biased conditioning and evidence inconsistency. In this setting, PPO is used as a practical optimizer for two coupled discrete actions under task-level feedback, rather than as the only viable choice.
>
> | Sel. | Det. AUC | Det. F1 | Loc. F1 | Loc. IoU |
> |---|---:|---:|---:|---:|
> | non-adap. | 0.958 | 0.944 | 0.660 | 0.552 |
> | Soft (non-RL) | 0.965 | 0.951 | 0.671 | 0.561 |
> | G-ST (non-RL) | 0.968 | 0.954 | 0.678 | 0.569 |
> | REINF (RL) | 0.970 | 0.957 | 0.682 | 0.575 |
> | PPO (RL) | **0.972** | **0.960** | **0.686** | **0.580** |
>
> Under the same state/action/reward design, PPO performs best among the tested optimizers. Thus, the core contribution is DAPO’s dual-action control, while PPO is a practical optimizer for it.
>
> **3. About Prompt Bank**
>
> We are sorry the manuscript did not detail the FPM prompt library enough. Evidence set $C$ contains 9 categories: generative traces, compression inconsistencies, resampling/interpolation artifacts, texture/statistical anomalies, boundary/blending irregularities, illumination/shadow inconsistencies, face-specific inconsistencies, text-rendering anomalies, and other contextual anomalies. Templates are fixed: *forensic evidence of* $C$, *image region with* $C$, and *possible manipulated area showing* $C$. Thus, Prompt Bank = 9 categories × 3 templates.
>
> These categories and templates were designed to cover common forensic evidence across forgery types while keeping the prompt space compact, controlled, and reproducible. Categories decide what to describe, while templates decide how to phrase it. Freezing $E_t$ is appropriate because the Prompt Bank uses short, coarse evidence descriptors rather than open-ended forensic jargon, keeping prompts in a stable semantic space and reducing prompt-side drift.
>
> **4. About training data configuration**
>
> Tables 1–2 use per-dataset training/testing rather than unified cross-dataset training. All reported results in Tables 1–2 are obtained on the test split, with a 7:1:2 train/val/test split for each dataset. To add cross-dataset evidence, we train on CASIA v1 + COVERAGE + NIST16 + Columbia and test directly on the unseen generative-editing domain CocoGlide, without target-domain fine-tuning and in bank-only mode.
>
> | Method | Det. AUC | Det. F1 | Loc. F1 | Loc. IoU |
> |---|---:|---:|---:|---:|
> | TruFor | 0.846 | 0.801 | 0.592 | 0.517 |
> | CoDE | 0.861 | 0.816 | 0.618 | 0.544 |
> | FPDA | **0.913** | **0.867** | **0.701** | **0.626** |
>
> **5. About image action space**
>
> $A_{\text{img}}=\{\text{skip},\text{strong},\text{edge}\}$ is designed for three common coarse-mask needs: conservative update, region completion/connectivity, and boundary refinement. We also test action composition and granularity:
>
> | $A_{\text{img}}$ | Det. AUC | Det. F1 | Loc. F1 | Loc. IoU |
> |---|---:|---:|---:|---:|
> | strong | 0.966 | 0.953 | 0.676 | 0.568 |
> | skip, strong | 0.969 | 0.956 | 0.681 | 0.574 |
> | skip, edge | 0.968 | 0.955 | 0.680 | 0.572 |
> | strong, edge | 0.970 | 0.958 | 0.683 | 0.576 |
> | skip, strong, edge | **0.972** | **0.960** | **0.686** | **0.580** |
> | skip, weak, strong, edge | 0.971 | 0.959 | 0.684 | 0.578 |
>
> Removing any mode hurts performance, while adding more granularity does not help. This suggests that $\{\text{skip},\text{strong},\text{edge}\}$ is a compact and effective action space.

---

> > ### Author Rebuttal · Reviewer_4ttY · 2026-04-01
> >
> > I truly appreciate the author's thoughtful responses and the additional experiments, which I believe have addressed some of the concerns. However, I still hold the view that the core contribution of the paper stems largely from the combination of numerous existing techniques, lacking deeper analytical insights—something currently needed in this field. I suggest the authors isolate a specific component of their method and provide thorough validation to demonstrate its effectiveness, as a strong paper should emphasize well-justified and verified contributions rather than an aggregation of multiple methods. Therefore, I maintain my original score. I again thank the authors for their responses and hope they can further improve the manuscript.

---

> > > ### Author Response · Authors · 2026-04-03
> > >
> > > We sincerely appreciate your further feedback. We understand your concern that the method may appear to be a combination of numerous existing techniques. In response to this point, we would like to state more explicitly that the central technical contribution of our method is the **dual-action mechanism for sample-adaptive evidence usage**.
> > >
> > > The key question addressed in our work extends beyond constructing a structured Prompt Bank. It concerns, for each input sample, **which forensic evidence should be activated and how that evidence should influence localization and refinement on the image side**. These two aspects correspond to the two actions in the dual-action mechanism:
> > >
> > > - **text-side action**: determines how forensic evidence in the prompt space should be routed and selected, i.e., *what evidence should be used for the current sample*;
> > > - **image-side action**: determines how that evidence should be coupled with image-side refinement, i.e., *how the selected evidence should take effect*.
> > >
> > > These two actions correspond to two complementary aspects of evidence usage: **evidence selection** and **mode of evidence application**. The former determines which forensic semantics the model should attend to, while the latter determines how those semantics enter the local visual representation and localization process. Taken together, they form a complete sample-level **evidence-to-localization pathway**. Accordingly, we view the most concentrated technical core of the paper as a **feedback-conditioned dual-action evidence usage mechanism**, in which prompt-side evidence routing and image-side refinement selection are jointly learned under final detection and localization feedback.
> > >
> > > Our ablation results further support this interpretation. To isolate this core mechanism more directly, we compared static evidence usage, non-RL adaptive selection, and the dual-action mechanism optimized by PPO under the same state, action, and reward setting:
> > >
> > > | Selector / Optimizer | Det. AUC | Det. F1 | Loc. F1 | Loc. IoU |
> > > |---|---:|---:|---:|---:|
> > > | Non-adaptive | 0.958 | 0.944 | 0.660 | 0.552 |
> > > | Soft (non-RL) | 0.965 | 0.951 | 0.671 | 0.561 |
> > > | G-ST (non-RL) | 0.968 | 0.954 | 0.678 | 0.569 |
> > > | REINF (RL) | 0.970 | 0.957 | 0.682 | 0.575 |
> > > | **PPO (RL)** | **0.972** | **0.960** | **0.686** | **0.580** |
> > >
> > > This trend indicates that the key performance gain comes from shifting evidence usage from a static regime to a sample-adaptive one; in our formulation, this adaptivity is realized through the **dual-action mechanism**. In other words, the component that we believe is most important to isolate and validate is not a particular optimizer itself, but this **dual-action evidence usage mechanism**.
> > >
> > > From this perspective, the other design elements in the paper are better understood as implementation support for this core mechanism in detection and localization. We hope this clarification helps further concentrate the contribution on the part of the method that is most appropriate to examine independently.
> > >
> > > Finally, we sincerely thank you again for your thoughtful and constructive comments throughout the review process. Your feedback has helped us identify and articulate the central technical contribution of the paper more clearly, and we will make this contribution statement more explicit in the camera-ready version.

---

### Official Review · Reviewer_1gza · 2026-03-12

**Soundness:** 3
**Presentation:** 4
**Significance:** 3
**Originality:** 3
**Overall Recommendation:** 4
**Confidence:** 4

**Summary:**

This paper proposes a novel joint image forgery detection and localization framework called FPDA. Its core contributions include: 1) a Forensic Prompting Module (FPM), which replaces open LLM cueing with a structured, reproducible cue library and a reliability gating mechanism for optional text input; and 2) a  Dual-Action Policy Optimization  (DAPO), which employs a one-step PPO algorithm to learn sample-adaptive discrete control for cue routing (a_txt) and mask refinement scheduling (a_img). The method is extensively evaluated on various datasets (CASIA, NIST16, CocoGlide, etc.), demonstrating superior or comparable performance to state-of-the-art methods in both detection and localization tasks.

**Compliance With Llm Reviewing Policy:**

Affirmed.

**Final Justification:**

The rebuttal resolves most of my concerns, and I will maintain my current recommendation.

**Key Questions For Authors:**

1. please provide a detailed list and sample template of the evidence category (c) used to build the FPM prompt library.
2. please explain the rationality of the design of the non labeled sample positioning reward (equation 14), and discuss the potential defects.
3. please quantify the robustness results (e.g. F1 score at a specific disturbance level) and report the additional training costs introduced by DAPO.
4. please evaluate the effect of the scheme on the new forgery based on GaN or diffusion.

**Limitations:**

yes

**Strengths And Weaknesses:**

Strength
1.  The problems identified with open-ended LLM prompting (irreproducibility, noise) and the need for sample-adaptive refinement are valid and clearly stated. FPM and DAPO are direct, novel solutions to these problems.
2. The protocol handling both text-available and text-missing settings under a single inference formulation is elegant and practical.
3. The results in Tables 1 & 2 are impressive.

Weakness
1. The process for constructing the structured prompt bank is described as "deterministic" based on an evidence category set C and templates. This is a black box. What exactly are the categories and templates? How were they chosen? This is a critical part of the method's prior knowledge.
2. While many baselines are included, some very recent and strong methods in the localization space (e.g., methods from CVPR 2024, ICCV 2024) are absent. The authors should justify their baseline selection timeline or include more recent comparisons.
3. In Table 2, many cells for baselines are marked as "-" (unavailable). For a fair comparison, it is crucial to either run those baselines under the same setting to fill the gaps or explicitly state that you are reporting numbers as cited from their original papers, acknowledging that evaluation protocols may differ. The current presentation is ambiguous.
4. The prompt bank is built on "common observable forensic evidence dimensions." How well would this bank generalize to a completely new type of forgery (e.g., Stable Inpainting)?

---

> ### Author Rebuttal · Authors · 2026-03-30
>
> We thank the reviewer for the thoughtful feedback.
>
> **1. About Prompt Bank**
>
> We are sorry that the manuscript did not describe the FPM prompt library clearly enough. The evidence category set $C$ contains 9 categories: (1) Generative traces; (2) Compression inconsistencies; (3) Resampling/interpolation artifacts; (4) Texture/statistical anomalies; (5) Boundary/blending irregularities; (6) Illumination/shadow inconsistencies; (7) Face-specific inconsistencies; (8) Text-rendering anomalies; and (9) Other contextual anomalies. The template family is not open-ended, but a small controlled set: (1) *forensic evidence of* $\{C\}$; (2) *image region with* $\{C\}$; and (3) *possible manipulated area showing* $\{C\}$. Thus, the Prompt Bank is a finite, reproducible library built from 9 categories and 3 templates. Example prompts include *forensic evidence of compression inconsistencies* and *image region with boundary/blending irregularities*.
>
> These categories were chosen to cover common observable forensic evidence dimensions across diverse forgery types. Categories determine what to describe, while templates determine how to phrase it. Thus, the Prompt Bank is not a black-box prompt source, but an explicit, evidence-centric, reproducible prior. We will present the 9 categories and template family in the camera-ready version.
>
> **2. About unseen generative forgeries**
>
> Thanks for your suggestion. To test transfer beyond the manual-tampering training family, we train on CASIA v1 + COVERAGE + NIST16 + Columbia and test directly on the unseen generative-editing domain CocoGlide, without target-domain fine-tuning and under the bank-only setting (no external text). Results follow.
>
> | Method | Det. AUC | Det. F1 | Loc. F1 | Loc. IoU |
> |---|---:|---:|---:|---:|
> | TruFor | 0.846 | 0.801 | 0.592 | 0.517 |
> | CoDE | 0.861 | 0.816 | 0.618 | 0.544 |
> | FPDA | **0.913** | **0.867** | **0.701** | **0.626** |
>
> FPDA achieves the best results on this unseen generative-editing domain across all four metrics. This supports the transferability of the scheme to new generative-editing forgeries, in our case diffusion-style, beyond the manual-tampering training family.
>
> **3. About baseline selection and fair comparison in Table 2**
>
> Thanks for your suggestion. We agree that fair comparison requires strong baselines and protocol consistency. Our baseline selection follows two principles: (1) **task relevance**, i.e., the method targets pixel-level forgery localization or joint detection/localization; and (2) **protocol comparability**, i.e., it can be compared directly under our benchmark-wise evaluation scope, valid-mask setting, and pixel-level F1/IoU.
>
> Under these criteria, Table 2 already includes recent strong baselines spanning CVPR’19--TPAMI’25, such as TruFor (CVPR’23), CoDE (TIFS’24), SAFIRE (AAAI’24), FakeShield (ICLR’25), and PIM (TPAMI’25). We also checked recent methods, but did not insert their published numbers into the main table when their dataset scope, evaluation setting, or localization metrics were not directly comparable to our Table 2 protocol, since doing so would mix protocols and weaken fairness.
>
> Accordingly, the symbol “–” means no directly comparable result is available, either because the method did not report results on the relevant dataset/setting, or because reliable re-evaluation is not possible due to unavailable code/models. Thus, the main table keeps a representative, protocol-compatible comparison set rather than an exhaustive mixed-protocol collection.
>
> **4. About non-labeled sample positioning reward (Equation 14)**
>
> Thanks for pointing this out. Our original intention in introducing the non-labeled sample positioning reward in Eq. (14) was to show that the reward can be extended to weakly supervised or partially annotated settings, where some samples may have image-level labels but no pixel-level annotations. However, this extended setting is not part of the current paper’s experimental protocol. We apologize for this ambiguity. In the revision, we will retain and emphasize only Eq. (13), and delete Eq. (14).
>
> **5. About quantified robustness results and training cost**
>
> Thanks for your suggestion. We further quantify the robustness trend in Figure 5 using representative severe disturbance levels. FPDA achieves pixel-level F1 of **0.70** at JPEG quality 50, **0.69** at Gaussian noise with standard deviation 13, **0.70** at Gaussian blur with kernel size 11, and **0.75** at resize rate 0.6; under the same levels, CoDE obtains **0.51 / 0.54 / 0.52 / 0.57**, and TruFor obtains **0.50 / 0.40 / 0.38 / 0.42**. This makes the robustness gap explicit and is consistent with the more gradual degradation trend already shown in Figure 5.
>
> Regarding the additional cost introduced by DAPO, comparing **+FPM → FPDA** in Appendix Table 7 shows only **+0.9M** parameters, **+0.8G** FLOPs, **+0.8 ms** latency, and **+0.2 GB** peak memory.

---

> > ### Author Rebuttal · Reviewer_1gza · 2026-04-02
> >
> > The rebuttal resolves most of my concerns, and I will maintain my current recommendation.

---

> > > ### Author Response · Authors · 2026-04-03
> > >
> > > Thank you very much for your follow-up and for your careful consideration of our rebuttal. We are encouraged that our response was able to resolve most of your concerns.
> > >
> > > We sincerely appreciate your thoughtful evaluation and constructive feedback throughout the review process. Thank you again for your time and careful reading.

---

### Official Review · Reviewer_ChTw · 2026-03-12

**Soundness:** 3
**Presentation:** 3
**Significance:** 3
**Originality:** 2
**Overall Recommendation:** 5
**Confidence:** 3

**Summary:**

This paper discusses the problem of image forgery detection, where modern editing and diffusion-based generation methods make forensic suggestions very weak and difficult to predict. Various existing methods depend on visual features only, while some recent approaches use vision–language models with textual prompts. However, independent prompts generated by large language models may cause noise and unstable direction during detection. To overcome this issue, the authors propose a framework called Forensic Prompting with Dual-Action Policy Optimization (FPDA). In this framework, a Forensic Prompting Module (FPM) is used to construct a structured set of forensic prompts that provide stable guidance to the model. The framework also allows optional textual input, but a reliable way is used so that unreliable prompts do not affect the detection process. In addition, a Dual-Action Policy Optimization (DAPO) strategy is applied to dynamically select the appropriate prompts and refinement strategy for each input image. This selective approach helps the model to focus on discriminative features and maintain a spatial group of features when generating forgery localization masks. The proposed approach is evaluated on various public datasets, including CASIA, NIST16, Coverage, CocoGlide, OpenForensics, and SID, which contain different generative scenarios such as diffusion-generated images and face forgeries. Experimental results show that the proposed method achieves better detection and localization performance compared with various existing methods.

**Compliance With Llm Reviewing Policy:**

Affirmed.

**Final Justification:**

The authors have addressed the comments and are in acceptable form to accept.

**Key Questions For Authors:**

Due to a number of heterogeneous components, the computational cost is high.
Clip Text and optional text do not discriminative mean that why it was adopted.
Equations and relevant text are not synchronized e.g., Et(t) in section 3.1.3. It is not sequentially understandable contextually.
Top-level feature C5 is not described.  Section   3.1.2
The evaluation loss function is not defined in this paper.
It is not clear whether prediction results are obtained from the training set or the test set.

**Limitations:**

yes

**Strengths And Weaknesses:**

The method introduces a structured forensic prompt bank that provides more controlled control to the vision–language prompts instead of random prompts.
It uses a dual-action policy optimization strategy that dynamically selects prompts and a refinement method that depends on the features of each input image.
The framework focuses not only on detection but also on localization and improves the spatial stability of the predicted forgery masks.
It supports different types of used images, including traditional editing, diffusion-based generation, and face forgery datasets.
The evaluation is performed on multiple public datasets, which demonstrate the effectiveness of the proposed method across different structured states.
The framework involves several components, such as the prompt bank and policy optimization module, which increase the overall model complexity.
The performance may depend on how well the forensic prompt bank is designed, and limited prompts may reduce the performance of the system.
The experiments are mainly conducted on benchmark datasets, so the generalization to real-world unseen manipulations is not fully clear.
The adaptive prompt and modification strategy may increase computational cost during inference.
The system relies on vision–language interaction, and noisy or unclear text prompts may still affect the detection stability in some cases.

---

> ### Author Rebuttal · Authors · 2026-03-30
>
> We thank the reviewer for the thoughtful feedback.
>
> **1. About complexity and computational cost**
>
> Thanks for this suggestion. FPDA adds components over a visual-only baseline, but the overhead is small and controlled. We evaluated the compute overhead on the Columbia dataset (Appendix Table 7). Compared with the visual-only baseline, full FPDA adds only **1.6M** parameters, **2.0G** FLOPs, **1.4 ms** latency, and **0.3 GB** peak memory.
>
> | Variant | Params (M) | FLOPs (G) | Latency (ms) | Peak Mem. (GB) |
> |---|---:|---:|---:|---:|
> | Visual-only | 46.2 | 158.0 | 27.8 | 8.1 |
> | +FPM | 46.9 | 159.2 | 28.4 | 8.2 |
> | +DAPO | 47.1 | 159.5 | 28.8 | 8.3 |
> | FPDA | 47.8 | 160.0 | 29.2 | 8.4 |
>
> **2. About Prompt Bank and optional text**
>
> Firstly, the Prompt Bank covers common forensic evidence types, including generative traces, compression inconsistencies, resampling artifacts, texture/statistical anomalies, and boundary irregularities. It is built as a deterministic prompt set from an evidence category set, a finite template family, and controlled slot enumeration, rather than open-ended prompt engineering. CLIP text is not used as an independent classifier, but as a stable shared semantic space for both the structured Prompt Bank and optional text.
>
> Secondly, optional text $t$ is only a reliability-aware cue. When text is missing or invalid, the model reduces to the deterministic bank-only condition $p_{\mathrm{FPM}}=p_{\mathrm{bank}}$; when text is present, the gate suppresses unreliable or noisy text rather than forcing its use. Text reliability is reflected by the fusion weight produced by the gate: given the structured bank prior and the text embedding, the gate assigns a smaller mixing coefficient to weakly aligned text, while missing or invalid text reduces to deterministic bank-only conditioning. Consistent with this design, ungated text slightly hurts performance, while gated text performs best in our ablation.
>
> Thirdly, to address “Prompt Bank quality / prompt count sensitivity,” we combine coverage ablation with the w/o Router comparison on SID Set description:
>
> | Coverage | Router | Det. AUC | Det. F1 | Loc. F1 | Loc. IoU |
> |---|:---:|---:|---:|---:|---:|
> | 25% | × | 0.986 | 0.971 | 0.663 | 0.573 |
> | 25% | ✓ | 0.988 | 0.973 | 0.668 | 0.577 |
> | 50% | × | 0.988 | 0.973 | 0.666 | 0.576 |
> | 50% | ✓ | 0.991 | 0.976 | 0.673 | 0.582 |
> | 100% | × | 0.990 | 0.975 | 0.670 | 0.580 |
> | 100% | ✓ | **0.993** | **0.979** | **0.679** | **0.588** |
>
> **3. About generalization performance**
>
> Thanks for this suggestion. We further add a generative evaluation experiment: the model is trained only on the manual-tampering datasets CASIA v1 + COVERAGE + NIST16 + Columbia, and tested directly on the unseen generative-editing domain CocoGlide, without target-domain fine-tuning and under the bank-only setting (no external text). The results are shown below.
>
> | Method | Det. AUC | Det. F1 | Loc. F1 | Loc. IoU |
> |---|---:|---:|---:|---:|
> | TruFor | 0.846 | 0.801 | 0.592 | 0.517 |
> | CoDE | 0.861 | 0.816 | 0.618 | 0.544 |
> | FPDA | **0.913** | **0.867** | **0.701** | **0.626** |
>
> **4. About clarity of exposition**
>
> **Regarding $E_t(t)$**: we agree Sec. 3.1.3 can be more linear. The current text does not explicitly unfold the text-conditioning path step by step, i.e., frozen text encoder $E_t(t)\rightarrow p_{\text{text}}\rightarrow \text{Gate}\rightarrow \eta_{\text{mix}}$. As a result, the role of $E_t(t)$ in the final conditioning formula is not sufficiently clear when first introduced. We will rewrite this part more explicitly and sequentially.
>
> **Regarding $C_5$**: we will add its definition where it first appears in Sec. 3.1.2: “$C_5$ denotes the top-level 1/32 feature map output by the Visual Encoder $E_v$; it is used to derive the global visual context $z_v$ for prompt routing and policy-state construction, and as the prompt-injection feature for RDA before multi-scale fusion.”
>
> **Regarding the evaluation loss function**: the DAPO reward is already defined in Sec. 3.2.3; what is not yet fully formalized in the main text is the joint supervised objective of the main network $\omega$. To avoid ambiguity, we will add its structural form in the revision, namely $L_{\text{sup}}=L_{\text{det}}(p,y)+\lambda_{\text{mask}}L_{\text{mask}}(\hat{M},M_{\text{gt}})$, and distinguish more clearly between the supervised optimization of $\omega$ and the PPO-based policy optimization of $(\theta,\phi)$.
>
> **Regarding training/testing data**: we are sorry for the confusion. All reported results are obtained on the test split. For all datasets, we use a 7:1:2 train/val/test split, where train is used for model learning, val for model selection and early stopping, and test for final evaluation. For baselines, we cite reported numbers only when the evaluation protocol is directly comparable; otherwise, we re-evaluate them under our setting.

---

> > ### Author Rebuttal · Reviewer_ChTw · 2026-04-03
> >
> > Plz decide based on the joint comments from all the reviewers.

---

> > > ### Author Response · Authors · 2026-04-04
> > >
> > > Thank you very much for your follow-up and for your careful consideration of our rebuttal. We are very glad that our response was able to fully address your concerns, and we sincerely appreciate your thoughtful evaluation throughout the review process.

---

### Decision · Program_Chairs · 2026-04-30

**Decision:**

Accept (regular)

**Comment:**

This paper proposes a joint image forgery detection and localization framework FPDA. The authors' rebuttal partially resolves the reviewers' concerns. However, multiple reviewers still have concerns regarding the novelty, analytical justification.

Despite the rebuttal, reviewers feel the paper aggregates existing techniques (e.g., structured prompting,  proximal policy optimization, specific fusion modules) without providing a sufficiently deep analysis of why this specific combination is superior beyond empirical gains. Reviewers view the work as a solid but incremental extension of existing paradigms in vision-language forensics, lacking the novel analytical insights or theoretical justification required to outweigh the introduced complexity. Therefore, this manuscript  is recommended for “Weak accept”.